# Changing Agendas on Sleep, Treatment and Learning in Epilepsy (CASTLE) Sleep-E: a protocol for a randomised controlled trial comparing an online behavioural sleep intervention with standard care in children with Rolandic epilepsy

Nadia Al-Najjar,[1] Lucy Bray [iD],[2] Bernie Carter [iD],[2] Advisory Panel CASTLE,[2] Amber Collingwood,[3] Georgia Cook,[4] Holly Crudgington [iD],[3] Janet Currier,[2] Kristina Charlotte Dietz [iD],[3] Will A S Hardy,[5] Harriet Hiscock,[6] Dyfrig Hughes [iD],[5] Christopher Morris [iD],[7] Deborah Roberts,[2] Alison Rouncefield-Swales [iD],[2] Holly Saron,[2] Catherine Spowart [iD],[1] Lucy Stibbs-Eaton,[1] Catrin Tudur Smith [iD],[8] Victoria Watson,[8] Liam Whittle [iD],[8] Luci Wiggs,[4] Eifiona Wood [iD],[5] Paul Gringras [iD],[9] Deb K Pal [iD] [3]

For numbered affiliations see end of article.

**Correspondence to**
Dr Kristina Charlotte Dietz;
kristina.dietz@kcl.ac.uk

## ABSTRACT

**Introduction** Sleep and epilepsy have an established bidirectional relationship yet only one randomised controlled clinical trial has assessed the effectiveness of behavioural sleep interventions for children with epilepsy. The intervention was successful, but was delivered via face-to-face educational sessions with parents, which are costly and non-scalable to population level. The Changing Agendas on Sleep, Treatment and Learning in Epilepsy (CASTLE) Sleep-E trial addresses this problem by comparing clinical and cost-effectiveness in children with Rolandic epilepsy between standard care (SC) and SC augmented with a novel, tailored parent-led CASTLE Online Sleep Intervention (COSI) that incorporates evidence-based behavioural components.

**Methods and analyses** CASTLE Sleep-E is a UK-based, multicentre, open-label, active concurrent control, randomised, parallel-group, pragmatic superiority trial. A total of 110 children with Rolandic epilepsy will be recruited in outpatient clinics and allocated 1:1 to SC or SC augmented with COSI (SC+COSI). Primary clinical outcome is parent-reported sleep problem score (Children's Sleep Habits Questionnaire). Primary health economic outcome is the incremental cost-effectiveness ratio (National Health Service and Personal Social Services perspective, Child Health Utility 9D Instrument). Parents and children (≥7 years) can opt into qualitative interviews and activities to share their experiences and perceptions of trial participation and managing sleep with Rolandic epilepsy.

**Ethics and dissemination** The CASTLE Sleep-E protocol was approved by the Health Research Authority East Midlands (HRA)–Nottingham 1 Research Ethics Committee (reference: 21/EM/0205). Trial results will be disseminated to scientific audiences, families, professional groups, managers, commissioners and policymakers. Pseudo-anonymised individual patient data will be made available after dissemination on reasonable request.

**Trial registration number** ISRCTN13202325.

## STRENGTHS AND LIMITATIONS OF THIS STUDY

⇒ First randomised controlled trial to evaluate the clinical and cost-effectiveness of a novel, tailored, parent-led Changing Agendas on Sleep, Treatment and Learning in Epilepsy (CASTLE) Online Sleep Intervention (COSI) that incorporates evidence-based behavioural components for children with Rolandic epilepsy.

⇒ Extensive patient and public involvement via dedicated CASTLE Advisory Panel.

⇒ Embedded health economic evaluation.

⇒ Heavily reliant on parent and child self-report to assess intervention implementation, ameliorated by COSI e-analytics and actigraphy data.

## INTRODUCTION

Epilepsy is one of the most common long-term neurological conditions worldwide whose prevalence peaks during childhood (5–9 years) and later in life (over 80 years).[1] Epilepsy in children (5–<13 years) accounts for the annual loss of 2.6 million disability-adjusted life years, equivalent to 1.8% of the global burden of disease among children and

adolescents.[2] Rolandic epilepsy (RE) is the most common childhood epilepsy.[3]

In the UK, RE has a stable crude incidence rate of 5 in 100 000 children (<16 years) or 542 new cases annually.[4] Concurrent neurodevelopmental disorders are very common (35%).[5] Seizures are often triggered by sleep fragmentation.[6] Many parents co-sleep or monitor children with nocturnal seizures, and children experience a fear of death during and after a seizure.[7] Problems related to sleep emerge as a top concern for both children and parents,[8] but are often unaddressed.[9 10]

A recent systematic review and meta-analysis of clinical trials shows that parent-based behavioural sleep interventions are effective for typically developing children and those with neurological and neurodevelopmental disorders.[10] The review concluded that randomised controlled clinical trials assessing functional outcomes (eg, cognition, emotion, behaviour) and targeting specific populations (eg, epilepsy) are missing (but see two recent trials).[11 12] Harms capture for cognitive–behavioural and behavioural sleep interventions has been sparse (only 32.3% of trials address adverse events (AEs)) and predominantly inadequate (92.9% of trials do not meet adequate reporting criteria).[13] Observed harms of behavioural sleep interventions in adults have been mild (eg, transient fatigue/exhaustion from sleep restriction in insomnia in 25%–33% of participants).[14] The only published paediatric and adult epilepsy trials did not address harms.[11 12] Based on the existing evidence, the benefits of behavioural sleep interventions in children with epilepsy outweigh potential harms, especially because sleep problems not only affect seizure control, but overall child well-being, learning and memory, and parental quality of life.[9 10] There remains, however, uncertainty whether sleep interventions, which can be resource intensive, are cost-effective in public health systems.

This protocol describes the design for the Changing Agendas on Sleep, Treatment and Learning in Epilepsy (CASTLE) Sleep-E trial, which evaluates the clinical and cost-effectiveness of a novel, tailored, parent-led CASTLE Online Sleep Intervention (COSI) that incorporates evidence-based behavioural components for children with epilepsy. COSI and CASTLE Sleep-E outcome selections were co-produced by affected children, young people and their parents, sleep and epilepsy experts.[8 15–17] The CASTLE Sleep-E protocol follows Standard Protocol Items: Recommendations for Interventional Trials (SPIRIT),[18 19] its extension for Patient-Reported Outcomes,[20] and the Guidance for Reporting Involvement of Patients and the Public (GRIPP2).[21]

As CASTLE Sleep-E is a pragmatic superiority trial assessing whether UK standard care (SC) for children with RE should be augmented with an online behavioural sleep intervention, SC is the appropriate comparator.[22–24] Current UK clinical guidelines[25–27] recommend that SC for children with RE consists of a comprehensive care plan with the option of pharmacological treatment with anti-epileptic drugs (AEDs).

The primary objective of CASTLE Sleep-E is to determine if SC augmented with COSI is superior to SC alone in reducing sleep problems in children with RE and cost-effective. Implementation details and secondary objectives are reported in table 1.

## METHODS AND ANALYSES
### Trial design
CASTLE Sleep-E is a UK-based, multicentre, open-label, active concurrent control, randomised (1:1), parallel-group, pragmatic superiority trial (overall trial start date: 14 May 2018, first trial site opened: 12 May 2022, first recruitment: 30 August 2022, planned trial end date: 31 July 2023). Compared are clinical and cost-effectiveness of SC alone and SC augmented with a novel, tailored, parent-led COSI (SC+COSI) in reducing sleep problems in children (5–<13 years) with RE at 3 and 6 months after randomisation. Parents and children (≥7 years) can opt into qualitative interviews and activities to share their experiences and perceptions within 3 weeks of completion of other data collection at 3 and 6 months after randomisation.

### Patient and public involvement
The CASTLE Programme (which subsumes CASTLE Sleep-E) recruited a dedicated patient and public involvement (PPI) Advisory Panel (AP) through social media and epilepsy charities in 2017. The CASTLE AP (CAP) consists of 17 adults with experience of childhood epilepsy and 5 children with epilepsy (aged 6–15 years). CAP has been involved in CASTLE from the funding application onward (two CAP members are co-applicants). Full PPI details are provided in GRIPP2-Short Form in table 2.

### Trial setting and eligibility criteria
Participants will be identified by staff in National Health Service (NHS) outpatient general paediatric and paediatric epilepsy clinics in the UK (predominantly urban setting). Eligibility criteria for participants are reported in online supplemental table 1, field 14 of the WHO Trial Registration Data Set (V.1.3.1). In the UK, a clinical RE diagnosis is based on electroclinical criteria defined by the International League Against Epilepsy (https://www.ilae.org/). Semiology and electroencephalogram (EEG) need to be judged as concordant by a consultant neurophysiologist. Neuroimaging does not form part of UK SC for RE. Eligibility criteria for trial sites include a capacity and capability assessment as advised for NHS site set-up by the UK Health Research Authority (HRA). The expected number of trial sites is 40 (England: 34, Scotland: 4, Wales: 1, Northern Ireland: 1). A list of trial sites can be obtained from the trial manager (see online supplemental table 1).

### Intervention
Participants will be allocated to trial arms (SC or SC+COSI) using minimisation (1:1 ratio). On allocation to SC+COSI, participants will receive an email with access

**Table 1** Outcomes for CASTLE Sleep-E (including participant-level metrics, time points, aggregation method)

| Outcome type | Specific measurement variable | Collected for | Participant-level analysis metric | Measurement time point(s) |
|---|---|---|---|---|
| **Primary** | | | | |
| 1. Clinical | Children's Sleep Habits Questionnaire[33] | Child | Total score | Baseline, 3 months |
| 2. Health economic | Cost utility of COSI*: National Health Service and Personal Social Services perspective, using outcomes 13–15 | Child and Parent | ▶ Time integral of utility<br>▶ Total costs | Baseline, 3 months, 6 months, (PLICS and HES at 6 months only) |
| **Secondary** | | | | |
| 1. Clinical | Children's Sleep Habits Questionnaire[33] | Child | Total score | Baseline, 6 months |
| 2. Clinical | Seizure-free period | Child | Time to first seizure from randomisation (days) | Randomisation, 3 months, 6 months |
| 3. Clinical | Seizure remission | Child | Time to 6-month seizure remission from randomisation (days) | |
| 4. Clinical | Knowledge about Sleep in Childhood (unpublished custom-scale) | Parent | Total score | Baseline, 3 months |
| 5. Clinical | Hospital Anxiety and Depression Scale[49] | Parent | Total score | Baseline, 3 months, 6 months |
| 6. Clinical | Insomnia Severity Index[43] | Parent | Total score | |
| 7. Clinical | SleepSuite[34] (iPad app) | Child | Reaction time (ms)<br>Executive function (accuracy) | Baseline, 3 months |
| 8. Clinical | ▶ Health-Related Quality Of Life Measure for Children with Epilepsy[32]<br>▶ WHO–Five Well-Being Index[50] | Child and parent | ▶ Total score<br>▶ Total score | Baseline, 6 months |
| 9. Clinical | Strengths and Difficulties Questionnaire[51] | Child | Total score | Baseline, 3 months, 6 months |
| 10. Clinical | Parenting Self-Agency Measure[52] | Parent | Total score | |
| 11. Clinical | Actigraphy[53] | Child and parent | ▶ Total sleep time (min)<br>▶ Sleep latency (min)<br>▶ Sleep efficiency (% asleep of sleep period)<br>*All 2-week averages* | Baseline, 3 months |
| 12. Clinical | Sickness-related school absences | Child | Total number of days | Randomisation, 3 months, 6 months |
| 13. Health economic | Health utilities derived from:<br>▶ EQ-5D-Y[40]<br>▶ Child Health Utility Instrument[39]<br>▶ EQ-5D-5L[54] | Child and parent<br>▶ Child<br>▶ Child<br>▶ Parent | Total score<br>▶ Utility score<br>▶ Utility score<br>▶ Utility score | Baseline, 3 months, 6 months |
| 14. Health economic | Insomnia Severity Index mapped to EQ-5D health state utilities[44] | Parent | Total score<br>▶ Utility score | Baseline, 3 months, 6 months |
| 15. Health economic | Direct costs: National Health Service and Personal Social Services perspective, measured using:<br>▶ Resource Use Questionnaire<br>▶ Case Report Form data<br>▶ PLICS data<br>▶ HES data<br>▶ Serious adverse events (assessed at 3 months, 6 months) | Child | Resource use and total cost | Baseline, 3 months, 6 months, (PLICS and HES at 6 months only) |
| 16. Health economic | Indirect and direct non-medical costs, measured using:<br>▶ Resource Use Questionnaire<br>▶ Case Report Form data | Child and parent | Resource use and total cost | Baseline, 3 months, 6 months |
| 17. Health economic | Cost utility of COSI: societal perspective, using quality-adjusted life years and cost using outcomes 13, 14 and 16 | Child and parent | ▶ Quality-adjusted life years from the time integral of utility<br>▶ Mean of total costs | Baseline, 3 months, 6 months |
| Qualitative | Trial experience | Child and parent | Qualitative interview transcript<br>Activity booklet transcript/photos | 3 months+3 weeks<br>6 months+3 weeks |

Child measures may be collected by parent proxy.
*Reported as incremental cost per quality-adjusted life year gained.
CASTLE, Changing Agendas on Sleep, Treatment and Learning in Epilepsy; COSI, CASTLE Online Sleep Intervention; HES, Hospital Episode Statistics; PLICS, Patient-Level Information and Costing Systems.

details to COSI. COSI consists of a self-paced, novel, tailored, e-learning package for parents of children with epilepsy that incorporates evidence-based behavioural components. Table 3 provides a brief overview; detailed reports on the development, content and evaluation of COSI have been published.[15 16] COSI is divided into 13 modules (1 screening for child-specific sleep problems to allow tailoring, 10 content, 1 additional resources, 1 initially hidden evaluation), of which 3 are compulsory (1 screening, 2 content). The non-compulsory modules are recommended based on screening outcome, but all modules are accessible, repeatable and printable. The

**Table 2** GRIPP2-Short Form[21] in research

| Section and topic | Item |
|---|---|
| 1: Aim<br>Report the aim of PPI in the study | To contribute to and guide the CASTLE Sleep-E study:<br>► To ensure greater relevance and acceptability of the study and study procedures to children with epilepsy and their parents.<br>► To ensure the study is communicated to families and the public in an accessible way (eg, recruitment, dissemination). |
| 2: Methods<br>Provide a clear description of the methods used for PPI in the study | Two adults with experience of childhood epilepsy are co-applicants on the CASTLE Research Programme National Institute for Health and Care Research (NIHR) Award (https://tinyurl.com/ycyfkc63) and are an integral part of the CASTLE Advisory Panel (CAP). CAP is a dedicated PPI Advisory Panel that was recruited in 2017 through social media and epilepsy charities. CAP consists of 17 adults with experience of childhood epilepsy and 5 children with epilepsy (aged 6–15 years). CAP members are reimbursed for expenses and offered honorarium payments in acknowledgement of their contributions. Facilitated by a salaried Family Engagement Officer and the PPI lead (LB), CAP members have co-developed working practices (CAP Handbook: adult version https://tinyurl.com/28u8jex4, child version: https://tinyurl.com/2p8d6bnx) and undertaken research training. CAP members communicate by video conference, telephone, email, social media and face-to-face. CAP is represented in the Trial Steering Group (see online supplemental table 2). CAP feedback and opinion are formally communicated to the CASTLE Sleep-E Trial Management Group (see online supplemental table 2) via the CASTLE PPI lead (LB). |
| 3: Study results<br>Outcomes—report the results of PPI in the study, including both positive and negative outcomes | To date (at the recruitment stage of CASTLE Sleep-E), CAP has contributed to the following trial aspects:<br>Initial funding application<br>Two adults with experience of childhood epilepsy are co-applicants on the CASTLE Research Programme NIHR Award (https://tinyurl.com/ycyfkc63).<br>Trial design<br>► CAP strongly endorsed the investigation focus (sleep problems) and the focus on non-seizure-related issues linked to epilepsy.<br>► CAP tested and consulted on the trial intervention (CASTLE Online Sleep Intervention) in respect to content, format and acceptability (eg, knowledge evaluation quiz was changed from compulsory to optional).<br>► CAP informed the selection of study questionnaires to ensure relevance to parents and children with epilepsy.<br>► CAP guided trial design to ensure acceptability of processes (eg, time, effort, schedule from a family perspective).<br>Trial procedure<br>► CAP led the development of a trial flow chart and clinician's guide (top tips for explaining the trial to families to aid recruitment).<br>► CAP guided data collection processes (assent/consent procedure, delivery of equipment, instructions, and packaging of actigraphs and iPads).<br>► CAP guided the qualitative interview content and format (eg, topics, question wording, length, delivery method and format).<br>Trial materials<br>► CAP informed the logo design (eg, CASTLE website: https://castlestudy.org.uk/) and name of the CASTLE Sleep-E trial.<br>► CAP guided the development of all participant-facing trial materials including):<br>Information Sheets and Consent Forms.<br>Child-friendly postcards to update and maintain interest in the trial.<br>Wording of trial emails sent to participating families, strap lines for promotional materials (eg, mugs and pens for trial sites).<br>Dissemination<br>► CAP informed liaison with stakeholders via social media and direct contact (charities, patient groups).<br>► CAP developed lay summaries for completed work as part of the CASTLE Programme and helped ensure the CASTLE Sleep-E trial website (https://castlesleepetrial.org.uk/) is accessible to families.<br>► CAP informed ongoing work to attract new CAP members. |
| 4: Discussion and conclusions<br>Outcomes—comment on the extent to which PPI influenced the study overall. Describe positive and negative outcomes | ► To date (recruitment stage of CASTLE Sleep-E), overall positive outcomes of CAP contributions to CASTLE Sleep-E have resulted in a trial design, procedure, materials and dissemination that is likely to have greater appeal and relevance to parents of children affected by Rolandic epilepsy and to the children themselves. CAP has made the trial more family focused, and enabled more direct public involvement (eg, contact details of the Family Engagement Officer on the CASTLE Sleep-E webpage). This should increase the proportion of eligible patients to assent/consent to trial participation. Materials (including the trial intervention itself) and procedures should be more accessible and more feasible to complete for participants, which should positively affect adherence, compliance and retention. Throughout their involvement, CAP contributions to the CASTLE Programme have exceeded expectations, and taken on a greater, independent purpose (eg, forming a support group via social media). The COVID-19 pandemic meant that CAP's work had to move online, and while this has facilitated engagement between CAP members across the country, it made it more difficult for the children to join in some of the consultation exercises. |
| 5: Reflections/critical perspective<br>Comment critically on the study, reflecting on the things that went well and those that did not, so others can learn from this experience | To be confirmed (currently at recruitment stage of CASTLE Sleep-E). |

CASTLE, Changing Agendas on Sleep, Treatment and Learning in Childhood Epilepsy; GRIPP2, Guidance for Reporting Involvement of Patients and the Public; PPI, patient and public involvement.

**Table 3** Content of the CASTLE Online Sleep Intervention (COSI)

| Module | Module name | Outline content | Compulsory or recommended |
|---|---|---|---|
| A | What is sleep and why is it important | Education about normal sleep physiology and processes | Compulsory |
| B | Sleep and seizures: a vicious cycle | Information about the relationship between sleep and seizures | Compulsory |
| C | Personalising this advice for your child | A sleep screening questionnaire to identify key areas of concern or problems around individual child sleep | Compulsory |
| D | Tips on sleep hygiene for everyone | General advice about key aspects of sleep hygiene | Recommended for all |
| E | Advanced sleep behaviour training | Introduction to principles of behavioural sleep interventions | Recommended for all |
| F | Learning difficulties, attention deficit hyperactivity disorder and autism spectrum disorders | Advice for parents of children with other comorbid conditions | Recommended to parents who highlighted (in module C) their child may have comorbid conditions |
| G | Solving falling asleep problems | Sleep intervention options for typical falling asleep problems | Recommended to parents who highlighted (in module C) their child may have problems falling asleep |
| H | Solving difficult night wakings and early morning waking | Behavioural techniques to address typical night or early waking problems | Recommended to parents who highlight (in module C) their child may have problems with their sleep during night or early morning wakings |
| I | Solving night-time fears | Behavioural techniques to address typical night-time fears | Recommended to parents who highlight (in module C) their child may have problems with night-time fears |
| J | Sleep walking, sleep terrors and nightmares | Information about different sleep behaviours, what causes them and how to identify and manage different conditions | Recommended to parents who highlight (in module C) their child may have problems with sleep walking, sleep terrors and/or nightmares |
| K | Troubleshooting and maintaining good sleep | How to deal with common issues, such as the child being ill or parents disagreeing about how to manage sleep and advice about how to maintain any benefits | Recommended to all |
| L | Resources | Links to additional resources of support, information and advice relating to sleep | Recommended to all |
| M | Evaluation | Questionnaire in which parents are asked to report on their experiences of using COSI | Recommended to all |

CASTLE, Changing Agendas on Sleep, Treatment and Learning in Epilepsy.

advice in COSI supports parents to implement general prevention techniques (eg, good sleep hygiene) and specific behavioural change techniques (eg, bedtime fading) relevant to their child's sleep problems. Three months after first being given access to COSI, parents will be asked by email to complete a COSI evaluation module. At the end of a participant's trial timeline (6 months), access to COSI will be revoked. After the trial, all families (irrespective of trial allocation) have the option to receive the COSI content in electronic format via email.

### Fidelity, adherence, retention and acceptability

Fidelity (intervention delivery) will be monitored through e-analytics embedded in the COSI system (modules accessed and time spent per module). Strategies to improve completion of COSI training in case of non-access include: (1) an automated text reminder after 2 days; (2) an email reminder after 4 days; (3) a phone call from researchers who developed COSI (the Sleep Team) after 6 days. To improve adherence to the intervention, (1) all participants will receive a phone call from the Sleep Team 6 weeks after account creation; and (2) children will receive postcards with child-oriented activities (eg, maze) at three time points to welcome them to the trial (weeks 1–2), to stay in touch (weeks 4–5) and to thank them for participating (weeks 4–8 post-trial).

To encourage completion of the intervention evaluation, participants will receive: (1) an automated text reminder after 3 days of non-completion, (2) and a phone call from the Sleep Team after 8 days of non-completion. Fidelity (intervention implementation, acceptability, perceived helpfulness) will be captured jointly by the COSI evaluation module and the qualitative trial component.

### Discontinuation, withdrawal, concomitant care or interventions

Participants may discontinue the trial intervention or withdraw from the trial if (1) the parent/child withdraws consent/assent, respectively; or (2) a change in the child's condition justifies discontinuation of treatment in their clinician's opinion. Trial site staff will record withdrawal with reason where provided in electronic Case Report Forms (eCRFs). Pseudo-anonymised data up to the time of consent withdrawal will be included in analyses in accordance with General Data Protection Regulation (GDPR)[28] under the UK Data Protection Act 2018[29]—the trial data controller relies on the legal bases of 'public interest' and 'research purposes'.

To avoid confounding and to minimise participant burden, co-enrolment into other clinical trials is discouraged. Where recruitment into another trial is considered appropriate, the trial coordinating centre will discuss

enrolment with the chief investigator (CI). Participation in the Rolandic Epilepsy Genomewide Association International Study (https://childhoodepilepsy.org/research-studies/regain/) is complementary (same CI).

### Outcomes and participant timeline

Outcomes are reported in table 1 and were chosen collaboratively by children and young people with epilepsy and their parents, sleep and epilepsy experts[8 17] in accordance with Core Outcome Measures in Effectiveness Trials guidelines.[30] Psychometric properties and clinical relevance of outcomes are reported in online supplemental table 3. Each participant will be followed up for 6 months. The participant timeline and estimated time requirement are, respectively, shown in table 4 and online supplemental table 4.

### Sample size

The target sample size (110 children with RE, 55 per trial arm) was calculated based on achieving 90% power

**Table 4** CASTLE Sleep-E participant timeline and order of outcome completion

| | T–4 weeks* Consent and baseline | T0† Randomisation | T+3 months Follow-up visit | T+6 months Follow-up visit |
|---|---|---|---|---|
| Visit no | 1 | 2 | 3 | 4 |
| Informed consent/assent | X | | | |
| Review of medical history and EEG results | X | | | |
| Eligibility confirmation | X | X | | |
| COVID-19 screener | X | | X | |
| Review of seizure occurrence | | X | X | X |
| Hospital admissions | | X | X | X |
| Demographics | X | | | |
| School absences | | X | X | X |
| Check contact details for accuracy | | X | X | X |
| Children's Sleep Habit Questionnaire[33] | X | | X | X |
| SleepSuite[34] (iPad) | X | | X | |
| WHO–Five Well-Being Index[50] | X | | | X |
| Health-Related Quality Of Life Measure for Children with Epilepsy[55] | X | | | X |
| Strengths and Difficulties Questionnaire[51] | X | | X | X |
| CHU-9D/CHU-9D proxy[39] | X | | X | X |
| EQ-5D-Y/EQ-5D-Y proxy[40] | X | | X | X |
| EQ-5D-5L[54] | X | | X | X |
| Parenting Self-Agency Measure[52] | X | | X | X |
| Insomnia Severity Index[43] | X | | X | X |
| Hospital Anxiety and Depression Scale[49] | X | | X | X |
| Resource Use Questionnaire | X | | X | X |
| Knowledge about Sleep in Childhood | X | | X | |
| Randomisation Standard care (SC) or (SC+COSI)[16] | | X | | |
| Intervention arm only: COSI[16] | | ←————————————→ | | |
| Actigraphy and sleep diary[53] (14 days) | X | | X | |
| Confirm continuing trial participation | | | X | X |
| Assessment of serious adverse events | | | X | X |
| Completion of follow-up Case Report Form | | | X | X |
| Review of concomitant medications | | X | X | X |
| Qualitative interview‡ | | | X | X |

*Up to 4 weeks flexibility between consent and randomisation to allow delivery of actigraph and iPad.
†Randomisation may be performed once 2 weeks of actigraphy and the minimum dataset are complete.
‡Optional trial component: consenting participants are interviewed within 3 weeks of follow-up visits 3 and 4.
CASTLE, Changing Agendas on Sleep, Treatment and Learning in Epilepsy; CHU-9D, Child Health Utility Index 9D; COSI, CASTLE Online Sleep Intervention; EEG, electroencephalogram.

to detect the minimal clinically important difference (MCID) in the primary clinical outcome (Children's Sleep Habits Questionnaire (CSHQ)) at 3 months after randomisation, accounting for 10% expected attrition (non-parametric test with two-sided 5% significance level). MCID was defined based on an individual-focused anchor-based method,[31] that is, 'the smallest difference in outcome that patients perceive as beneficial and which mandates a change in patient management'.[32] The MCID value was based on the estimated reduction in total CSHQ score required for children with epilepsy (M=48.25, SD=8.91)[7] to fall at or below the diagnostic cut-off score of 41 for sleep disorders in paediatric populations.[33]

### Recruitment, stopping guidelines and interim analyses

An Independent Data and Safety Monitoring Committee will monitor recruitment and make recommendations to the Trial Steering Committee (TSC) concerning trial continuation, adjustments of recruitment methods and follow-up optimisation (see online supplemental table 2). A traffic light approach will determine trial continuation: (1) green: continue if at least 30 trial sites have opened and 22 participants have been randomised by end of month 6; (2) amber: implement additional recruitment strategies if 15–21 participants have been randomised by end of month 6; (3) red: if recruitment is <15 participants by end of month 6, then stopping the trial early will be discussed with the TSC. Formal interim analyses of the accumulating data will not be performed.

### Treatment allocation

Participants will be allocated with a 1:1 ratio to either SC or SC+COSI based on a computer-generated adaptive restricted randomisation procedure that minimises differences between trial arms in variables likely to affect outcomes. Minimisation algorithm details are not published to avoid subversion of allocation sequence concealment, but include seizure frequency, AED and sleep medication details. The allocation concealment mechanism is an online, central randomisation service implemented and maintained by the Liverpool Clinical Trial Centre (LCTC). The service will be accessed within 4 weeks of participant enrolment (once consent and eligibility confirmed, participant ID issued, baseline dataset completed) by trained, authorised staff at trial sites. Randomisation will trigger allocation emails to the trial manager at LCTC and to the relevant trial site as well as enable COSI access for participants allocated to the intervention arm. Trial sites will notify the participant's general practitioner of the treatment allocation by letter (electronic or hard copy, depending on preference).

### BLINDING

Only quantitative data analysts will be blinded (participant IDs do not reveal treatment allocation). All other stakeholders (participants, parents, healthcare providers, data collectors, qualitative researchers) will be aware of the allocated intervention. Emergency unblinding procedures are therefore unnecessary.

### Assent and consent

Potentially eligible children will be screened at trial centres by trained site staff. Screening outcome will be documented. Eligible children with interested parents will be invited to participate and provided with a Patient Information Sheet and Consent (PISC) Form electronically and/or hard copy (PISC, three versions: parent, child (5–6 and 7–12 years)). Sufficient time will be allowed for discussion of the trial and the decision to assent/consent to trial entry and the optional qualitative component. Assent (children aged 7–12 years) and consent (parents) may be given face-to-face or remotely and will be electronically captured in a secure Consent Database managed by LCTC. Reasons for declining participation will be asked, but it will be made clear that children and parents do not have to provide a reason.

### Data collection and management

Data collection will be carried out electronically except for Serious Adverse Events and Participant Transfer Forms (hard copy). At consent/assent, site staff will enter patient medical history (including EEG), eligibility confirmation, COVID-19 screening and demographics (see table 4) into eCRFs stored in a secure Data Management System managed by LCTC. Trial participation will be added to the patient's medical records alongside their unique participant ID.

Consent and Contacts Databases are securely linked. The addition of a new participant will trigger email notifications to the parents containing access links to baseline assessments (see table 4) and the Sleep Team who will access the Contacts Database to arrange the delivery of an iPad preconfigured by LCTC (optionally fitted with prepaid SIMs), and two actigraphs with supporting documents. iPads (Generations 7–8, iOS V.15.2 or V.15.3) will be used to access the SleepSuite App (V.1.4),[34] which assesses executive functions in child-friendly, interactive games (eg, popping virtual bubbles with smiling children's faces). Access requires the participant ID and is only possible at prespecified trial time points (see table 4). Data are only stored on the iPad until the test session completion, then automatically uploaded to a cloud-based server, and then securely downloaded for analyses by authorised LCTC staff. Families lacking other means of internet access can use iPads fitted with prepaid SIMs to access other online trial materials (including email).

Actigraphs (Micro Motionlogger Watch and Watchware Software V.1.99.17.4, Ambulatory Monitoring, New York, USA) will be used to collect 14 days of objective sleep data from child and parent. Concurrent sleep diaries (hard copies) will be completed by the parent with or without child input. At the end of the baseline period, actigraphs will be returned to the Sleep Team via prepaid courier. The Sleep Team will download and securely store pseudo-anonymised (using participant IDs) actigraphy data for

pre-processing (manual selection of sleep periods cross-checked against sleep diaries) per night at participant level. Summary variables (sleep latency, total sleep time and sleep efficiency) are then automatically calculated by actigraph software, manually collated and securely transferred electronically to LCTC for trial-level analyses by the trial statistician.

Participants will be randomised to trial arms during a telephone/video call or clinic visit only *after* site staff have confirmed that baseline data (see table 4) are complete, and eligibility, consent/assent and contact details are still valid. Data collection will be repeated 3 and 6 months after randomisation, and iPads to LCTC via trial sites (see table 4).

The Qualitative Research Team will access the Contacts Database to schedule audio-recorded interviews with children and parents who consented/assented to this optional trial component. Interviews (audio or audio-video) will take place remotely within 3 weeks of completion of other data collection at 3 and 6 months after randomisation. Parents and children will be interviewed together or separately as preferred. Parents and children will have the opportunity to think through their ideas prior to the interview (as proposed by parents and children from the CAP). Children will be invited to complete activity booklets in advance of their interviews (the booklets will be mailed or emailed 1 week prior to their interview); the content they complete will support the interview. Parents will receive a list of proposed questions/topics. Children will be able to share the booklet with the Qualitative Team (eg, screen or photograph sharing, verbal description).

The direct costs of health and personal social services, and indirect costs of productivity losses and school absenteeism will be collected using a Resource Use Questionnaire administered at baseline and during follow-up visits. Other data such as concomitant medications, study visits and AEs will be collected using eCRFs. Trial participants' use of secondary care services will be collected from Patient-Level Information and Costing Systems (PLICS) data obtained from the finance departments of each recruiting hospital or from Hospital Episode Statistics (HES) data obtained from NHS Digital at the end of the trial. PLICS and HES data will be pseudo-anonymised and transferred securely to the trial health economists at Bangor University.

## Data quality, security and trial oversight

Reliability, validity and clinical relevance of outcomes are reported in online supplemental table 3. Processes to promote quality and security of collected data include general local training of site staff and research teams (Good Clinical Practice), and trial-specific training in the use of electronic forms and databases by LCTC. LCTC will request to see evidence of appropriate training and experience of all trial staff. Staff will be signed off as appropriately qualified by the CI. Electronic data capture provides several in-built validity and security checks (eg, data type, range and missingness checks in eCRFs,

SleepSuite use/access restrictions). Some electronic and all hard copy data will be repeat checked (eg, eligibility, contact details). Data processing requiring more subjective judgement will be performed by minimum of two trained researchers on at least a subset of data (ie, manually assisted selection of actigraphy sleep period; thematic and content analysis of qualitative data).

Data will be processed and stored in accordance with GDPR under the UK Data Protection Act 2018. Central data monitoring will be performed by LCTC which will raise and resolve queries with site and research teams within the online system. The University of Liverpool is registered with the Information Commissioners Office. LCTC will receive trial participants' HES identifiers for secure transfer to the Health Economic Team, who will access, securely store and dispose of HES data in accordance with the Bangor University and NHS Digital Data Sharing Framework Contract.

## Statistical methods

Statistical analyses of all but health economic and qualitative data will be performed by the trial statistician (LCTC) using SAS software, V.9.4 or later. Intention-to-treat will be the main analysis strategy for primary and secondary outcomes (see table 1 and table 5). Minimisation variables (including seizure frequency, AED and sleep medication details) will be adjusted for at baseline. Statistical significance will be set at the conventional two-sided 5% level; clinical relevance will be based on previous research (see online supplemental table 3). Point estimates with 95% two-sided CIs will be reported adjusted and unadjusted for covariates. No multiplicity adjustments will be made (only one primary clinical outcome, uncorrected secondary outcome analyses).

Sensitivity analyses will be carried out if the amount of missing data is greater than 10%. Multiple imputation will be used to assess the robustness of the analysis to missing primary outcome data. The multiple imputation method will follow published guidelines.[35] PROC MI in SAS (version 9.4 or later) will be used to generate 50 complete datasets. The imputation model will include all variables included in the primary outcome analysis model. The overall summary adjusted mean difference will be presented with 95% CIs, to assess the sensitivity of the primary analysis to missing data. All analyses will be reported in accordance with the Consolidated Standards of Reporting Trials Checklist[36] and regardless of statistical significance.

## Health economic evaluation

The economic analysis will be performed in accordance with a Health Economics Analysis Plan, and by the trial health economists at Bangor University. The primary analysis will adopt an NHS and Personal Social Services perspective and, based on quality-adjusted life years (QALYs) as a measure of health outcome, estimate the incremental cost-effectiveness ratio from an incremental analysis of the mean costs and QALYs for the intervention

**Table 5** Analysis plan for outcome variables in CASTLE Sleep-E (further analyses details are reported in text)

| Outcome type | Specific measurement variable | Hypothesis | Method of analysis |
|---|---|---|---|
| **Primary** | | | |
| Clinical | Children's Sleep Habits Questionnaire[33] | Total score lower in intervention arm at 3 months | Linear mixed effect regression:<br>▶ Fixed effects: intervention (binary)<br>▶ Random effects: trial site (categorical)<br>▶ Covariates:<br>Baseline score<br>Use of sleep medication (binary) |
| Health economic | Cost* per quality-adjusted life year gained | Not applicable (health economic evaluation) | Cost-effectiveness (utility) analysis |
| **Secondary** | | | |
| Clinical | Children's Sleep Habits Questionnaire[33] | Total score lower in intervention arm at 6 months | Linear mixed effect regression (as before) |
| Clinical | Seizure-free period | Time to first seizure (days) differs between trial arms at 3 and 6 months | Survival analyses<br>▶ Kaplan-Meier curves by trial arm<br>▶ Cox proportional hazards regression (if applicable)<br>– Covariates:<br>– Use of sleep medication (binary)<br>– Trial site (categorical) |
| Clinical | Time to 6-month seizure remission from randomisation (days) | Time to 6-month seizure remission (days) differs between trial arms at 6 months | Survival analyses (as before) |
| Clinical | ▶ Knowledge about Sleep in Childhood<br>▶ Actigraphy[53] (2-week average):<br>Total sleep time<br>Sleep latency<br>Sleep efficiency | Total score differs between trial arms at 3 months | Linear mixed-effects regression (as before) |
| Clinical | ▶ Hospital Anxiety and Depression Scale[49]<br>▶ Insomnia Severity Index[43] | Total score lower in intervention arm at 3 and 6 months | Linear mixed-effects regression (as before) |
| Clinical | ▶ Sickness-related school absences | Total days differ between trial arms at 3 and 6 months | Poisson mixed-effects regression |
| Clinical | ▶ Health-Related Quality Of Life Measure for Children with Epilepsy[55]<br>▶ WHO–Five Well-Being Index[50] | Total score differs between trial arms at 6 months | Linear mixed-effects regression (as before) |
| Clinical | ▶ SleepSuite[34]: animal task<br>▶ SleepSuite: bubble task<br>Shape detection<br>Emotion detection<br>Gender detection<br>▶ SleepSuite: maze task | Executive function, reaction time and variability differ between trials arm at 3 months | ▶ Poisson/zero-inflated negative binomial regression (depending on presence of overdispersion)<br>▶ 2×2 multivariate repeated-measures analysis of variance<br>Factors: Time (PM/AM)×intervention (pre/post)<br>Fitted per detection task (shape, emotion, gender)<br>▶ Linear mixed-effects regression (as before) |
| Clinical | ▶ Strengths and Difficulties Questionnaire[51]<br>▶ Parenting Self-Agency Measure[52] | Total score differs between trial arms at 3 and 6 months | Linear mixed-effects regression (as before) |
| Qualitative | Trial experience† | Not applicable (inductive) | Thematic analysis (interpretive, reflexive and conceptual analytical approach)<br>▶ Discrete sets: intervention/control, child/parent, engagement with intervention/lack thereof, decision-making types, responses/experiences<br>▶ Separately for child and parent, then jointly (dyad)<br>▶ Comparisons with selective objective data as emerging from analysis (eg, anxiety measures, actigraphy) |

*Perspective: NHS and PSS perspective; alternative perspective: societal (indirect and direct non-medical costs).
†Source data for trial experience: qualitative interviews (parents and children individually and as dyad), activity booklets (children only).
CASTLE, Changing Agendas on Sleep, Treatment and Learning in Epilepsy; NHS, National Health Service; PSS, Personal Social Services.

and control trial arms.[37] Data assumed to be missing at random will be imputed using multiple imputation by chained equations.[38]

Sensitivity analyses will be conducted to test whether, and to what extent, the incremental cost-effectiveness ratio is sensitive to key assumptions in the analysis (eg, unit prices, different utility estimates from Child Health Utility Index 9D[39] vs EQ-5D-Y[40]). The joint uncertainty in

costs and QALYs will be addressed through application of bootstrapping and estimation of cost-effectiveness acceptability curves.[41] Alternative scenarios considering a broader cost perspective (including indirect costs, such as school absences and loss of productivity, valued by reference to published sources) and a range of outcomes (including parental QALYs, measured using the EQ-5D-5L[42] and Insomnia Severity Index[43 44]) will be

conducted. Inclusion of spillover disutility[45] (impact on parents' utility) will be based on the National Institute for Health and Care Excellence reference case specification[46] that all QALYs are of equal weight and calculated assuming additive effects. Health economic findings will be reported according to the Consolidated Health Economic Evaluation Reporting Standards.[47]

### Qualitative component

Child and parent interviews will be analysed by the Qualitative Research Team using an interpretive, reflexive and conceptual analytical approach. Audio-recordings of interviews will be transcribed and thematically analysed in discrete sets (eg, intervention/control, child/parent, engagement/lack of engagement with intervention, types of decision-making, different responses/experiences). Parent and child transcripts will first be analysed separately, and then as dyads. All data will be used for synthesis. Thematic and content analyses will be used for child activity booklets (text and images). Qualitative and selected quantitative data (eg, anxiety measures, actigraphy data) will be compared, as appropriate.

### Harms

A flow chart of AE-reporting requirements is shown in online supplemental figure 1. Harms severity and causality will be graded by the investigator responsible for the care of the participant based on categories shown in online supplemental table 5. If any doubt about causality exists, the local investigator should inform LCTC who will notify the CI. In case of discrepant views, the Research Ethics Committee (REC) will be informed of both views. Seriousness and expectedness of AEs will be defined based on International Council for Harmonisation of Technical Requirements for Pharmaceuticals for Human Use Definitions and Standards for Expedited Reporting (ICH E2A, ref: CPMP/ICH/377/95). Expectedness will be assessed by the CI. The only expected AEs in CASTLE Sleep-E are mild and transient worsening of sleep behaviours targeted by the trial intervention. Safety data will be quality checked by a statistician not otherwise involved in the trial. Safety analysis will include all patients randomised and starting treatment and be presented descriptively split by treatment arm.

### Auditing

The CI will ensure that the trial team conducts monitoring activities of sufficient quality and quantity (eg, protocol adherence, consent/assent, data quality). The sponsor will delegate monitoring duties and activities to LCTC. The CI and LCTC will inform the sponsor of any concerns. Auditing does not meet the National Institute for Health and Care Research or SPIRIT statement definitions of independence[19 48] as auditors (LCTC and CI) are part of the trial team.

### Protocol amendments

Substantive protocol amendments will be notified to HRA via the UK's Integrated Research Application System. Trial sites will receive an amendment pack of HRA-approved and REC-approved changes and unless an objection is received within 35 days, the trial will continue at site with a GO LIVE email.

### Ancillary and post-trial care

King's College London (KCL) holds insurance against claims from participants for harm caused by their participation in this clinical study; compensation can be claimed in case of KCL negligence.

### Ethics and dissemination

The CASTLE Sleep-E protocol was approved by the HRA East Midlands–Nottingham 1 REC (reference: 21/EM/0205). Trial results will be disseminated to scientific audiences in peer-reviewed publications and conferences, and—with the help of the CAP (parent and child experts by experience), relevant charities (eg, Epilepsy Action, Epilepsy Society and Cerebra) and professional groups (eg, Royal College of Paediatrics and Child Health, Epilepsy Specialist Nurses Association)—as plain language summaries to families, other professional groups, managers, commissioners and policymakers. Pseudo-anonymised individual patient data and associated documentation (eg, protocol, statistical analysis plan, annotated blank CRF) will be made available after dissemination on reasonable request.

### Registration details

ISRCTN registry (trial ID: ISRCTN13202325, prospective registration: 09 September 2021). The WHO Trial Registration Data Set (V.1.3.1) for CASTLE Sleep-E is shown in online supplemental table 1.

**Author affiliations**

[1]Liverpool Clinical Trials Centre, Institute of Population Health, Faculty of Health and Life Sciences, University of Liverpool, Liverpool, UK

[2]Department of Nursing & Midwifery, Faculty of Health, Social Care and Medicine, Edge Hill University, Ormskirk, UK

[3]Department of Basic and Clinical Neurosciences, Institute of Psychiatry, Psychology & Neuroscience, King's College London, London, UK

[4]Centre for Psychological Research, Faculty of Health and Life Sciences, Oxford Brookes University, Oxford, UK

[5]Centre for Health Economics and Medicines Evaluation, School of Medical and Health Sciences, Bangor University, Bangor, UK

[6]Centre for Community Child Health, Murdoch Children's Research Institute, Parkville, Victoria, Australia

[7]University of Exeter Medical School, Faculty of Health and Life Sciences, University of Exeter, Exeter, UK

[8]Department of Health Data Science, Institute of Population Health, Faculty of Health and Life Sciences, University of Liverpool, Liverpool, UK

[9]Department of Sleep Medicine, Evelina London Children's Hospital, London, UK

**Collaborators** Nadia Al-Najjar, Lucy Bray, Bernie Carter, CAP, Amber Collingwood, Georgia Cook, Holly Crudgington, Jant Currier, Kristina C Dietz, Will A S Hardy, Harriet Hiscock, Dyfrig Hughes, Christopher Morris, Deborah Roberts, Alison Rouncefield-Swales, Holly Saron, Catherine Spowart, Lucy Stibbs-Eaton, Catrin Tudur Smith, Victoria Watson, Liam Whittle, Luci Wiggs, Eifiona Wood, Paul Gringras, Deb K Pal. CASTLE Advisory Panel (CAP): Steve Carr, Alisa Carr, Charlotte Carr, James Carr, Matthew Carr, Keith Hardy, Ned Hardy, Tim Ingram, Clare Ingram, Oliver Ingram, Rachael Martin, Charlotte Myer, Beccy Pile, Kelly Reynolds, Deborah Roberts, Rachel Roberts, Caris Stoller, Caroline Towell, Bethany Towell.

**Contributors** PG and DKP (chief investigators), CT-S, DR, HH, JC, LW, BC, CM, DAH and LB (co-investigators) conceived the study and are award holders. Topic expertise for the core outcome set development was provided by CAP, LB, BC, AC, HC, PG, DAH, CM, DKP, and CT-S. Epilepsy expertise by experience was provided by CAP. Topic expertise for epilepsy was provided by DKP. Topic expertise for the health economic evaluation was provided by WASH, DAH and EW. Topic expertise for intervention development was provided by GC, PG, HH, DKP and LW. Topic expertise for patient and public involvement (Advisory Panel and Family Engagement) was provided by CAP, AR-S, LB, BC and CM. Responsibility for the selection of patient-reported outcomes lay with CM. Responsibility for programme management lay with AC. Topic expertise for qualitative research components was provided by CAP, LB, BC and HS. Topic expertise for sleep was provided by GC, PG, HH and LW. Topic expertise for statistical analyses was provided by CT-S, VW and LWh. Responsibility for trial management lay with NA-N, CS and LS-E. All authors contributed to the design and refinement of the study protocol. The protocol manuscript was written by KCD (including supplemental materials but excluding figure 1 and Patient Information Sheet and Consent Forms). Authors in the Trial Management Group (TMG) had the opportunity to provide feedback twice (initial and final draft); non-TMG authors had the opportunity to provide feedback once (final draft). Provided feedback was incorporated. The final manuscript was approved for publication by all authors. GRIPP2 content was checked for accuracy by LB. Sponsor name and contact information are provided in online supplemental table 1. Details of trial committees and other groups and individuals overseeing the trial are listed in online supplemental table 2. Trial site principal investigators will be listed alphabetically in publications of trial results as members of the CASTLE Sleep-E Consortium in the Acknowledgements section. There has not been and will not be any use of hired writers.

**Funding** This work is supported by the National Institute for Health and Care Research (NIHR) (award number RP-PG-0615-20007). HH was supported by a National Health and Medical Research Council (NHMRC, Australia) Practitioner Fellowship (1136222). HH's institute—the Murdoch Children's Research Institute (MCRI, Australia)—is supported by the Victorian Government's Operational Infrastructure Support Program (no award/grant number).

**Competing interests** None declared.

**Patient and public involvement** Patients and/or the public were involved in the design, or conduct, or reporting, or dissemination plans of this research. Refer to the Methods section for further details.

**Patient consent for publication** Not required.

**Provenance and peer review** Not commissioned; externally peer reviewed.

**ORCID iDs**
Lucy Bray http://orcid.org/0000-0001-8414-3233
Bernie Carter http://orcid.org/0000-0001-5226-9878
Holly Crudgington http://orcid.org/0000-0003-1048-4953
Kristina Charlotte Dietz http://orcid.org/0000-0002-3074-6319
Dyfrig Hughes http://orcid.org/0000-0001-8247-7459
Christopher Morris http://orcid.org/0000-0002-9916-507X
Alison Rouncefield-Swales http://orcid.org/0000-0001-9947-7375
Catherine Spowart http://orcid.org/0000-0001-8641-2871
Lucy Stibbs-Eaton http://orcid.org/0000-0002-3672-4006
Catrin Tudur Smith http://orcid.org/0000-0003-3051-1445
Liam Whittle http://orcid.org/0000-0001-8280-1984
Eifiona Wood http://orcid.org/0000-0002-2785-7325
Paul Gringras http://orcid.org/0000-0002-0495-3517
Deb K Pal http://orcid.org/0000-0003-2655-0564

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
