## [Reviewer comments · BMJ Open]

ARTICLE DETAILS

TITLE (PROVISIONAL)	Changing Agendas on Sleep, Treatment and Learning in Epilepsy (CASTLE) Sleep-E: A protocol for a randomised controlled trial comparing an online behavioural sleep intervention with standard care in children with Rolandic epilepsy
AUTHORS	Group, CASTLE; Advisory Panel, CASTLE; Al-Najjar, Nadia; Bray, Lucy; Carter, Bernie; Collingwood, Amber; Cook, Georgia; Crudgington, Holly; Dietz, Kristina; Gringras, Paul; Hardy, Will A. S.; Hiscock, Harriet; Hughes, Dyfrig; Morris, Christopher; Pal, Deb; Rouncefield-Swales, Alison; Saron, Holly; Spowart, Catherine; Stibbs-Eaton, Lucy; Tudur-Smith, Catrin; Watson, Victoria; Whittle, Liam; Wiggs, Luci; Williamson, Paula; Wood, Eifiona

VERSION 1 – REVIEW

REVIEWER	Mifsud, Janet University of Malta, Clinical Pharmacology
REVIEW RETURNED	11-Aug-2022

GENERAL COMMENTS	This article is a well written describing a protocol for a randomised controlled trial comparing an online behavioural sleep intervention with standard care in children with Rolandic epilepsy. The paper clearly described the thorough methodology, tools and indicators to be used, the inclusion and exclusion criteria, the tools to be developed, timeline of interventions, ethical issues involved and involvement of patient organisations. The limitations are acknowledged as being heavily reliant as self reporting. It clearly sets the scene and explains the importance of such a study in this type of childhood epilepsy. Yet, at this stage, it is not clear as to why this protocol is being forwarded for publication when no results (not even pilot results) are as yet available. The protocol is particularly time intensive and demanding of the parents or parents proxy eg training, filling in of the various numerous questionnaires and tools, number of visits, distance from hospital etc as outlined in Table 4. This may be a limiting factor which would hinder the participation of children and reduce numbers in the trial since it will rely on the parents willingness or ability to contribute to the study. It may be necessary to quantify this time and ensure that the parent/proxies are well aware upfront of the time needed to ensure successful completion of the study. The level of education of the parents also needs to be considered, in order to assess their capacity to recall accurately their child's measurement. This will also greatly affect their level of
---

	understanding of the Content of the CASTLE Online Sleep Intervention (COSI) (Table 3). This has not been addressed in sufficient detail The type of drug or therapy the child may be on is also an important variable to consider as well as the frequency of seizures. This should also be considered in the analysis of the results. At this stage, it may be somewhat premature to submit this paper for publication, prior to at least having some pilot data. The results of such a pilot may greatly impact the practical aspects of the workings of such as study as it will give a better overview of how such a complex protocol will work in practice. Such a pilot would also give a better overview of the limitations of this complex protocol and identify and pre-empt problems and issues that may arise in the actual implementation of the larger trial.
--	---

REVIEWER	Lee, James British Columbia Children's Hospital
REVIEW RETURNED	31-Aug-2022

GENERAL COMMENTS	Well thought-out and written protocol that has potential to provide significant new insight into the effect of sleep intervention on a common epilepsy syndrome, in which sleep is highly overlooked. Could the authors provide more information about the eligibility criteria, specifically in terms of how diagnosis of RE will be made? For example, I presume it will be based on seizure semiology and EEG, but what about patients found to have rolandic EEG trait incidentally (I presume these would not be included but did not see this stated). Are there certain features of the EEG that would be required? For instance, independent bilateral discharges would be more reassuring that it is definitely RE, but unilateral discharges alone certainly do not exclude it by any means. Finally, what about imaging to confirm that it is not symptomatic of a structural lesion and really rolandic? Generally this is not performed in most patients with suspected RE but would it be considered to more strictly define the population for this study? How will different seizure frequencies among patients be handled, in terms of both eligibility and randomization? Some patients may have very few seizures, others very frequent. Similarly, how will AED treatment effects be accounted for in analysis (presumably, at the time of randomization the patients will not be on any medications yet, though I do not think I know that for certain, so it would not factor into randomization) since AEDs can affect sleep architecture? This is important since standard of care includes no treatment in most cases, but also would include AED treatment in those with more frequent episodes.
--

VERSION 1 – AUTHOR RESPONSE

Reviewer: 1

Comments to the Author:

Yet, at this stage, it is not clear as to why this protocol is being forwarded for publication when no results (not even pilot results) are as yet available.

[...]

At this stage, it may be somewhat premature to submit this paper for publication, prior to at least having some pilot data. The results of such a pilot may greatly impact the practical aspects of the workings of such as study as it will give a better overview of how such a complex protocol will work in practice. Such a pilot would also give a better overview of the limitations of this complex protocol and identify and pre-empt problems and issues that may arise in the actual implementation of the larger trial.

[...]

The protocol is particularly time intensive and demanding of the parents or parents proxy eg training, filling in of the various numerous questionnaires and tools, number of visits, distance from hospital etc as outlined in Table 4. This may be a limiting factor which would hinder the participation of children and reduce numbers in the trial since it will rely on the parents willingness or ability to contribute to the study. It may be necessary to quantify this time and ensure that the parent/proxies are well aware upfront of the time needed to ensure successful completion of the study.

*Response

- The publication type for this submission is ‘protocol manuscript’, which — by definition (e.g. BMJ Open: see <https://bmjopen.bmj.com/pages/authors#protocol>) — reports planned studies or studies at an early stage when results are not yet available. Registered reports are increasingly common across all disciplines to encourage good research practice (see <https://www.cos.io/initiatives/registered-reports>) and are seen as best practice for clinical trials alongside prospective registration.
- The trial procedure is complex and time-consuming especially from an administrative perspective but has been streamlined as much as possible from a participant perspective. The protocol has been piloted and approved in terms of both time and effort from a family perspective by the CASTLE Advisory Panel (CAP). CAP is a dedicated Patient and Public Involvement (PPI) Advisory Panel that consists of 17 adults with experience of childhood epilepsy and five children with epilepsy (aged 6–15 years).
- The estimated time-requirement of CASTLE Sleep-E participation varies between 2–3 hours per month over a 6-month period depending on trial arm and participation in optional qualitative interviews. Supplemental Table 3 has been added with a detailed break-down of how time-estimates are calculated. Whilst a time investment of 2–3 hours per months over a 6-months period is non-trivial, the CASTLE Sleep-E protocol was not judged as excessively burdensome by the CASTLE Advisory Panel (CAP). Participants provide informed consent and are made aware of trial requirements as well as their right to withdraw without providing any reason at any point. It would have been a good idea to include the estimated time-requirement in the trial protocol and trial information, but the trial is now live, and any modification of trial material requires formal approval. If there should be other reasons for formal amendments of the protocol, we will consider adding overall time estimates by trial arm to the Patient Information Sheet and Consent Form.

Reviewer: 1

Comments to the Author:

The level of education of the parents also needs to be considered, to assess their capacity to recall accurately their child’s measurement. This will also greatly affect their level of understanding of the Content of the CASTLE Online Sleep Intervention (COSI) (Table 3). This has not been addressed in sufficient detail

*Response

Parental education level is not formally captured for the CASTLE Sleep-E dataset, and analyses can therefore not be adjusted for this variable. Recruiting site staff will only assess whether families fulfil eligibility criteria (which do not require a specific parental education level).

As CASTLE Sleep-E is a pragmatic superiority trial assessing whether UK standard care for children with RE should be augmented with an online behavioural sleep intervention (COSI), it would not be appropriate to exclude parents based on their education level (COSI needs to work for any parental education level to be a viable augmentation to standard care in the UK).

Reviewer: 1

Comments to the Author:

The type of drug or therapy the child may be on is also an important variable to consider as well as the frequency of seizures. This should also be considered in the analysis of the results.

*Response

Details about anti-epileptic drug (AED) treatment as well as sleep medication (medication name, indication, administration route, dose, frequency, start date, end date) are collected at baseline, randomisation, and 3- and 6-month follow-up. We do not publish details of the minimisation algorithm used at the randomisation stage to avoid subversion of allocation sequence concealment. Sleep medication is included as a baseline variable for the analyses of primary and secondary outcomes. Parallel adjustments are made for AEDs where possible (depending on the proportion of participants in receipt of AEDs, and on number of participants per AED type, e.g. in respect of effects on sleep architecture).

Reviewer: 2

Comments to the Author:

Could the authors provide more information about the eligibility criteria, specifically in terms of how diagnosis of RE will be made? For example, I presume it will be based on seizure semiology and EEG, but what about patients found to have rolandic EEG trait incidentally (I presume these would not be included but did not see this stated). Are there certain features of the EEG that would be required? For instance, independent bilateral discharges would be more reassuring that it is definitely RE, but unilateral discharges alone certainly do not exclude it by any means. Finally, what about imaging to confirm that it is not symptomatic of a structural lesion and really rolandic? Generally this is not performed in most patients with suspected RE but would it be considered to more strictly define the population for this study?

* Response

The diagnosis of RE is made clinically using a combination of electroclinical criteria as described in the International League Against Epilepsy definitions (<https://www.ilae.org/>). The semiology and EEG need to be concordant to make the epilepsy syndrome diagnosis. A rolandic EEG trait without the corresponding seizure semiology would not meet eligibility criteria. The features of the EEG in rolandic epilepsy are variable between individuals and across time. The EEG features need to be judged consistent by a consultant neurophysiologist and all EEG reports will be checked for consistency. Neuroimaging is not considered standard of care for rolandic epilepsy in the UK.

Reviewer: 2

Comments to the Author:

How will different seizure frequencies among patients be handled, in terms of both eligibility and randomization? Some patients may have very few seizures, others very frequent.

* Response

Eligibility criteria do not preclude participation in CASTLE Sleep-E based on seizure frequency. We do not publish details of the minimisation algorithm used at the randomisation stage to avoid subversion of allocation sequence concealment.

Reviewer: 2

Comments to the Author:

Similarly, how will AED treatment effects be accounted for in analysis (presumably, at the time of randomization the patients will not be on any medications yet, though I do not think I know that for certain, so it would not factor into randomization) since AEDs can affect sleep architecture? This is important since standard of care includes no treatment in most cases, but also would include AED treatment in those with more frequent episodes.

* Response

Eligibility criteria do not preclude participation in CASTLE Sleep-E based on medication (including anti-epileptic drugs [AEDs]). Details about anti-epileptic drug (AED) treatment as well as sleep medication (medication name, indication, administration route, dose, frequency, start date, end date) are collected at baseline, randomisation, and 3- and 6-month follow-up. We do not publish details of the minimisation algorithm used at the randomisation stage to avoid subversion of allocation sequence concealment. Sleep medication is included as a baseline variable for the analyses of primary and secondary outcomes. Parallel adjustments are made for AEDs where possible (depending on the proportion of participants in receipt of AEDs, and on number of participants per AED type, e.g. in respect of effects on sleep architecture).

VERSION 2 – REVIEW

REVIEWER	Mifsud, Janet University of Malta, Clinical Pharmacology
REVIEW RETURNED	29-Nov-2022
GENERAL COMMENTS	The authors have provided extensive replies to the queries made by the reviewers and have edited the article accordingly. It is now acceptable for publication